# Complex Physical Structure of Complete Mitochondrial Genome of *Quercus acutissima* (Fagaceae): A Significant Energy Plant

**DOI:** 10.3390/genes13081321

**Published:** 2022-07-24

**Authors:** Dan Liu, Haili Guo, Jingle Zhu, Kai Qu, Ying Chen, Yingtian Guo, Ping Ding, Haiping Yang, Ting Xu, Qi Jing, Shangjun Han, Wei Li, Boqiang Tong

**Affiliations:** 1National Engineering Laboratory of Tree Breeding, Key Laboratory of Genetics and Breeding in Forest Trees and Ornamental Plants of Ministry of Education, The Tree and Ornamental Plant Breeding and Biotechnology Laboratory of National Forestry and Grassland Administration, College of Biological Sciences and Technology, Beijing Forestry University, Beijing 100083, China; 1821618@163.com (D.L.); qukai@bjfu.edu.cn (K.Q.); 2Shandong Provincial Center of Forest and Grass Germplasm Resources, Jinan 250102, China; ghaili0415@163.com (H.G.); dp0416@shandong.cn (P.D.); yanghaiping197911@shandong.cn (H.Y.); xuting0226@163.com (T.X.); jqgzh163@163.com (Q.J.); hsj3180044@163.com (S.H.); 3Research Institute of Non-Timber Forestry, Chinese Academy of Forestry, Zhengzhou 450003, China; zhujingle1982@126.com; 4Forestry Protection and Development Service Center of Shandong Province, Jinan 250109, China; chenying2509@163.com; 5College of Agriculture and Forestry, Linyi University, Linyi 276000, China; gytxiaozi@163.com

**Keywords:** *Quercus acutissima*, mitochondrial genome, repeated sequences, genome recombination, phylogenetic relationship

## Abstract

*Quercus acutissima* Carruth. is a Chinese important energy plant with high ecological and economic values. While the species chloroplast genome has been reported, its mitochondrial genome (mitogenome) is still unexplored. Here, we assembled and annotated the *Q. acutissima* mitogenome, and we compared its characteristic differences with several closely related species. The *Q. acutissima* mitogenome’s main structure is branched with three distinguished contigs (linear molecule 1, circular molecule 2, and circular molecule 3) with 448,982 bp total length and 45.72% GC content. The mitogenome contained 51 genes, including 32 protein-coding, 16 tRNA and 3 rRNA genes. We examined codon usage, repeated sequences, genome recombination, chloroplast to mitochondrion DNA transformation, RNA editing, and synteny in the *Q. acutissima* mitogenome. Phylogenetic trees based on 29 species mitogenomes clarified the species classification. Our results provided comprehensive information of *Q. acutissima* mitogenome, and they are expected to provide valuable information for Fagaceae evolutionary biology and to promote the species germplasm utilization.

## 1. Introduction

*Q**. acutissima*, Fagaceae, is a deciduous tree and is one of the three members of the East Asian branch of *Quercus* [1]. The species is widely distributed in China’s warm temperate and subtropical regions [2], which is characterized by fast growth, strong sprouting ability, early fast-growing period, drought resistance, and less strict soil requirements. The species is suitable for mountainous areas, hills and hillocks, growing mixed with either evergreen or deciduous trees and often is an upper layer species [3]. *Q. acutissima* forests have great production potential and can be used as timber and fuel (i.e., high economic value). Additionally, the species has high capacity for water and soil conservation, which are important ecological attributes [4]. The species has high resistance and absorption ability to sulfur dioxide, chlorine, hydrogen fluoride, and it is more resistant to fire and smoke. It is suitable for urban landscape, windbreak, fire prevention, and water connotation forests [5].

The mitochondria are important sites for energy synthesis and conversion for various cell physiological activities [6,7]. Mitochondria plays a crucial role in plant growth and development [8] as it converts biomass energy into chemical energy by phosphorylation, and it is involved in cell division, differentiation, and apoptosis [9,10,11]. Generally, for angiosperms, the nuclear genome is bi-parental inherited, while the chloroplasts and mitochondria, it is uniparentally inherited [12], thus eliminating paternal line influence and facilitating genetic mechanisms studies [13]. The complex nature of mitogenomes made it more difficult to assemble compared to other organelles genomes [14], as evident by the number of released organelles genomes (7427 chloroplast genomes, 449 mitogenomes, and 1120 plastid genomes: https://www.ncbi.nlm.nih.gov/genome/browse#!/organelles/, accessed on 4 May 2022).

Angiosperms mitogenomes vary widely in size and structure, ranging from 66 Kb (*Viscum scurruloideum* Barlow) [15] to 11.3 Mb (*Silene conica* L.) [16] with 19 to 64 [15] known genes (not including duplicate genes and ORFs), 5 to 25 [15,17] introns, and highly variable intergenic regions [18]. The conformation of plant mitogenomes is diverse, and most of the assembled mitogenomes are circular. However, some may be polycyclic, such as maize [19] and kiwifruit [20], or linear, such as *Lactuca sativa* L. [21], and some have multi-branched structures, such as Sitka spruce (*Picea sitchensis* (Bong.) Carrière) [22] mitogenome.

*Q. acutissima* is an excellent germplasm material for studying stepped terrain effects and paleoclimate change on the species evolution, genetic structure, population dynamics, and distribution history in China on a large spatial scale [23]. The cpDNA sequence-based analysis showed slightly higher genetic diversity and limited chloroplast gene flow (seed flow) in the *Q. acutissima* population [24]. Although the *Q. acutissima* chloroplast genome has been reported [25], the mitogenomes is not. Here, we assembled and annotated *Q. acutissima* mitogenome. This study aimed to analyze relative synonymous codon usage (RSCU) and repeated sequences, detect genome recombination, assess gene transfer between chloroplast and mitochondrial genomes, and RNA editing sites, and explore synteny and phylogenetic relationships. Our results are expected to provide a theoretical basis for species identification and biological research, they are of great significance for exploring the species origin and its evolutionary relationship, and they are expected to promote molecular systematics and conservation genetics research and applications.

## 2. Materials and Methods

### 2.1. Plant Materials and Sequencing

In May 2021, live *Q. acutissima* leaves were collected from the Aishan National Forest Park, Songshan street, Qixia City, Shandong Province (N 37°2′51′′ E 120°47′33′′). Plant specimens (barcode number sdf1001223) and total genomic DNA were stored in Shandong Provincial Center of Forest and Grass Germplasm Resources (Biao Han, hanbiao3361@shandong.cn, code htq2021cp10), and we obtained total DNA according to the steps of the Blood/Cell/Tissue Genomic DNA Extraction Kit (TIANamp Genoic DNA Kit) of Tiangen company. We used both the Nanopore GridION sequencing platform (Oxford Nanopore Technology, Oxford Science Park) and Illumina Novaseq 6000 platform to sequence and construct the library, obtaining raw sequence data (Nanopore raw data were 14.79 Gb, N50 were 9816 bp and Illumina raw data were 10.56 Gb).

### 2.2. Genome Assembly and Annotation

We assembled the *Q. acutissima* mitogenome using a combined strategy of Illumina and Nanopore. The second-generation DNA sequencing data were assembled using the default parameters of GetOrganelle v1.7.5 [26] to obtain a graphical mitogenome. The mitogenome was visualized using Bandage [27], and the single extended fragments of chloroplast and nuclear genome were manually removed. Then, the Nanopore data were aligned to the graphed mitogenome fragments using bwa software [28], and the resulting Nanopore data were used to resolve the repeated sequence regions of the graphed mitogenome. The final result is a branched multigenomeric structure. *Arabidopsis thaliana* (L.) Heynh. was selected as the reference genome for protein-coding genes (PCGs) of mitogenome, and Geseq [29] was used to annotate the mitogenome. The tRNA and rRNA of mitogenome were annotated using tRNAscan-SE [30] and BLASTN [31], respectively. Mitogenome annotation errors were manually corrected using Apollo [32]. The annotated mitogenome has been deposited into GenBank under accession number MZ636519.

### 2.3. Analysis of RSCU and Repeated Sequences

The protein-coding sequences of the genome were extracted using Phylosuite [33]. Mega 7.0 [34] was used to conduct codon preference analysis for PCGs of mitogenome and calculate RSCU values. MISA (https://webblast.ipk-gatersleben.de/misa/, accessed on 30 April 2022) [35], TRF (https://tandem.bu.edu/trf/trf.unix.help.html, accessed on 30 April 2022) [36], and REPuter web server (https://bibiserv.cebitec.uni-bielefeld.de/reputer/, accessed on 30 April 2022) [37] identified repeated sequences including simple sequence repeat (SSR), tandem repeat and interspersed repeat. The results were visualized using the Circos package [38].

### 2.4. Detection of Genome Recombination

BLASTN [31] was used to detect the repeated sequences in the *Q. acutissima* mitogenome, and a total of 160 results were obtained. Subsequently, 87 pairs of repeated sequences were obtained after manual exclusion. Then, two repeat units of the repeated sequences and their flanking 1000 bp regions were extracted as the primary conformation. Afterwards, the 1000 bp region upstream and downstream of the repeat unit was exchanged to artificially simulate the secondary conformation that could result from recombination. Finally, the three-generation data were mapped to these sequences of major and minor conformations, and by counting the number of long reads that completely span the repeated sequences, we determine whether there is a recombination of the genome.

### 2.5. Chloroplast to Mitochondrion DNA Transformation and RNA Editing Prediction

The chloroplast genome was assembled and annotated using GetOrganelle [26] and CPGAVAS2 [39], respectively. Homologous fragments were analyzed using BLASTN [31], and the results were visualized using the Circos package [38]. The prediction of RNA editing events was based on the online website PREP suit (http://prep.unl.edu/, accessed on 5 May 2022) [40].

### 2.6. Synteny and Phylogenetic Analysis

A dot plot of pairwise comparison was generated and plotted conserved co-linear blocks. Based on sequence similarity, a Multiple Synteny Plot of the *Q. acutissima* mitogenome with closely related species was plotted using MCscanX [41]. The mitogenome of closely related species were selected and downloaded (https://www.ncbi.nlm.nih.gov/, accessed on 5 May 2022) based on their affinity, and then, PhyloSuite [33] was used to extract shared genes. MAFFT [42] with a bootstrap value of 1000 was used for multiple sequence alignment analysis, and MRBAYES [43] was used for phylogenetic analysis. The results of the phylogenetic analysis were visualized in ITOL software [44].

## 3. Results

### 3.1. Q. acutissima Mitogenome Features

The main structure of the *Q. acutissima* mitogenome is branched, and after excluding duplicated regions from the Nanopore data, we obtained three contigs (molecules 1–3) with 448,982 bp total length and 45.72% GC content (Figure 1). The lengths of linear molecule 1, circular molecule 2, and circular molecule 3 were 224,233, 188,259, and 36,490 bp, respectively, and the GC contents were 45.88, 45.68, and 44.98%, respectively (Figure 2).

The *Q. acutissima* mitogenome was annotated with 32 PCGs (including 24 unique mitochondrial core and 8 non-core genes), 16 tRNA genes (*trn*E-UUC, *trn*M-CAU, and *trn*P-UGG are multi-copy), and 3 rRNA genes (Appendix A). Among the 32 PCGs, 10 contained introns (*rpl*2, *cox*2, *nad*1, *rps*3, *ccm*FC, *rps*10 and *nad*5 had a single intron, *nad*4 contained three introns, *nad*2 and *nad*7 had four introns), while two genes (*nad*1 and *nad*5) were trans-spliced. The core genes include five ATP synthase genes (*atp*1, *atp*4, *atp*6, *atp*8, and *atp*9), nine NADH dehydrogenase genes (*nad*1, *nad*2, *nad*3, *nad*4, *nad*4L, *nad*5, *nad*6, *nad*7, and *nad*9), four ubiquinol cytochrome c reductase genes (*ccm*B, *ccm*C, *ccm*Fc, and *ccm*Fn), three cytochrome c oxidase genes (*cox*1, *cox*2, and *cox*3), one transport membrane protein gene (*mtt*B), one maturases gene (*mat*R), and one cytochrome c biogenesis gene (*cob*). Non-core genes include three large subunit of ribosome genes (*rpl*2, *rpl*5, and *rpl*10), four small subunit of ribosome genes (*rps*3, *rps*4, *rps*10, and *rps*12), and one succinate dehydrogenase gene (*sdh*3).

### 3.2. PCGs Codon Usage Analysis

The codon usage analysis of 32 PCGs was performed, and the codon usage of each amino acid is shown in Appendix A. Codons (RSCU > 1) were considered to be used preferentially by amino acids. As shown in Figure 3, in addition to the RSCU values of 1 for both the start codon AUG (Met) and UGG (Trp), there is also a general codon usage preference for the mitochondrial PCGs. GCU (Ala), UAU (Tyr), and CAU (His) were the three most frequent codons in *Q. acutissima*.

### 3.3. Q. acutissima Mitogenome Repeats Analysis

A total of 3, 79, 68, and 11 SSRs were found in molecule 1, 2, and 3, respectively (Figure 4), and SSRs in monomeric and dimeric forms accounted for 41.77, 44.12, and 72.73% of the total SSRs, respectively. In molecule 1, thymine (T) monomeric repeats accounted for 52.94 % (9) of the 17 monomeric SSRs, TA repeats accounted for 31.25 % of the dimeric SSRs, and there were no hexameric SSRs. In addition, there were 12 tandem repeats with a match greater than 67% and length between 13 and 22 bp, and 58 pairs of interspersed repeats with length greater than or equal to 30 were observed, including 30 pairs of direct repeats (the longest being 216 bp) and 28 pairs of palindromic repeats. In molecule 2, thymine (T) monomeric repeats accounted for 43.75% (7) of the 16 monomeric SSRs, AT repeats accounted for 35.71% of the dimeric SSRs, and there were no hexameric SSRs. In addition, there were 16 tandem repeats with matches greater than 70% and lengths between 13 and 30 bp, and a total of 65 pairs of interspersed repeats with lengths greater than or equal to 30 were observed, including 48 pairs of direct repeats (the longest being 112 bp) and 17 pairs of palindromic repeats. In molecule 3, adenine (A) monomeric repeats accounted for 60.00% (3) of the 5 monomeric SSRs, and there were no trimeric, pentameric or hexameric SSRs. In addition, there were three tandem repeats with a match greater than 81% and a length between 18 and 22 bp, and a total of seven pairs of interspersed repeats with a length greater than or equal to 30 were observed, including four pairs of direct repeats (the longest being 35 bp) and three pairs of palindromic repeats.

### 3.4. Detection of Genome Recombination

There are not many repeated sequences in the *Q. acutissima* mitogenome, most of which are short in length, and no evidence of recombination was detected. However, evidence of recombination was detected for two pairs of longer repeats (Table 1). The length of the repeated sequence R1 is 10,578 bp; due to the limited length of ONT data, only 19 and eight long reads were detected to support the primary and secondary conformations, respectively. The two units of R1 are located on the molecule 1 (Figure 5), which allows the region between the two repeat units to be inverted. Another pair of repeated sequences R2 is 1679 bp in length, and due to the relatively short length, 224 and 182 long reads were detected to support the primary and secondary conformations, respectively. This indicates a high frequency of recombination occurring in this region. The R2 can mediate the formation of independent circular molecule 3 or integration of molecule 3 onto molecule1, but the proportion of the former is higher.

### 3.5. Chloroplast to Mitochondrion DNA Transformation

According to the sequence similarity analysis, a total of 23 fragments were homologous to the mitogenome and chloroplast genome, with a total length of 15,688 bp, accounting for 3.49% of the total length of the mitogenome (Figure 6, Table 2). Among them, four fragments exceeded 1000 bp, with fragments 1 and 2 being the longest at 4760 bp. By annotating these homologous sequences, 13 complete genes were identified, including 2 PCGs (*pet*L, *pet*G), 10 tRNA genes (*trn*V-GAC, *trn*I-GAU, *trn*A-UGC, *trn*D-GUC, *trn*M-CAU, *trn*I-CAU, *trn*W-CCA, *trn*P-UGG, *trn*N-GUU, *trn*H-GUG), and 1 rRNA gene (*rrn*16S).

### 3.6. The Prediction of RNA Editing

A total of 466 potential RNA editing sites were identified on 32 mitochondrial PCGs based on the online website PREP suit for the prediction of RNA editing events at a cutoff value = 0.2 criterion, all of which were C-T(U) edits (Figure 7). The predicted RNA editing sites in each gene are shown in Figure 7. On the mitochondrial genes, 38 RNA editing sites were identified for both *ccm*FN and *nad*4 genes, which were the most among all genes. The next highest number was 32 for *ccm*B.

### 3.7. Synteny and Phylogenetic Analysis

As shown in Figure 8, a positive repeat sequence of approximately 10 kb in length was identified in the *Q. acutissima* mitogenome by dot-plot analysis. The largest co-linear blocks of nearly 20 kb were identified in the dot plot with *Fagus sy**lvatica* L., and larger co-linear blocks were identified in the dot plot with *Juglans mandshurica* Maxim. and *A**. thaliana*, but they were smaller than 20 kb. A large number of homologous co-linear blocks were detected between *Q. acutissima* and the closely related species (Figure 9). The results indicate that the co-linear blocks are not arranged in the same order among individual mitogenomes; that is, the *Q. acutissima* mitogenome has undergone extensive genomic rearrangements with closely related species, and the mitogenomes is extremely unconserved in structure.

Due to the genome’s low substitution rate, mitochondrial genes are a valuable source of information for phylogenetic analysis at a high taxonomic level [45]. In order to determine the phylogenetic position of *Q. acutissima*, we used the mitogenome sequences of 29 angiosperm species from GenBank based on the sequences of 14 conserved mitochondrial PCGs to construct a phylogenetic tree, with *A**. thaliana* as the outgroup (Figure 10). Based on a relatively high support rate and in line with the latest classification of APG (Angiosperm Phylogeny Group), *Q. acutissima* belonged to the family Fagaceae of the order Fagales and is closely related to *F**. sylvatica*.

## 4. Discussion

Mitochondria are important organelles in eukaryotic cells, providing energy for various cells physiological activities. In fact, plant mitogenomes are more complex than those of animals, with extensive size variation, sequence alignment, repetitive content, and highly conserved coding sequences [12,21], and many mitogenome sequences of plants have been reported [46,47,48]. Although plant mitogenomes are often assembled and displayed as circular maps, plant mitochondrial DNA does, most likely, not exist as one large circular DNA molecule but mostly as a complex and dynamic collection of linear DNA with combinations of smaller circular and branched DNA molecules [21,49,50,51,52]. In the entire order of the Fagales, the assembled mitogenome sequences of three species have been made public: *Quercus variabilis* Blume (GenBank MN199236, unverified) [53], *Betula pendula* Roth. (GenBank LT855379.1, not annotated) [54], and *F**. sylvatica* (GenBank NC050960.1) [55]. The total *Q. acutissima* mitogenome length of 448,982 bp is between the *Q**. variabilis* (same family, same genus) (412,886 bp) and the *F**. sylvatica* (same family, different genus) (504,715 bp), while *B**. pendula* (same order, different family) is the longest (581,505 bp). The GC content is an important factor in the assessment of species [56], the GC content of the *Q. acutissima* mitogenome was 45.72%, which was close to the GC content of the *Q**. variabilis* mitogenome (45.76%) and *F**. sylvatica* mitogenome (45.8%). The *F**. sylvatica* mitogenome may fit best in a circular display, which is not corresponding to the physical structure of the genome in vivo, where it is more likely to exist in different conformations [55]. In the present study, the *Q. acutissima* mitogenome is a branched structure of two circular molecules and one linear molecule; however, the coexistence of these molecules needs to be further investigated.

There are 64 codons in the eukaryotic genome, and there is a wide variation in the rate of genomic codon usage among different species and organisms. This preference is thought to be the result of a relative equilibrium that gradually develops within the cell over a long period of evolutionary selection. In *Q. acutissima*, most PCGs were the typical ATG start codon, and the distribution of amino acid compositions was similar to other angiosperms [57,58].

Repeated sequences are widely present in the mitogenome and can be divided into two main categories: tandem and interspersed repeats [59]. Repeated sequences in the mitogenome are often critical for intermolecular recombination, and in general, the largest repeats within a species (usually more than about 1 kb in angiosperms) have been found to recombine constitutively, leading to isomerization [51,60]. The longest interspersed repeat sequence in the *Q. acutissima* mitogenome exceeded 1 kb (10,578 bp in size) and may be responsible for heterodimerization. In comparison, the longest repeat sequence of *Q. variabilis* mitogenome is 17.3 kb in size, the longest *F**. sylvatica* repeat is 918 bp, while the longest repeat existed in *B**. pendula* is only 474 bp. 

Homologous recombination mediated by repeats is almost universal in plant mitogenomes [61,62]. It has been reported that the size of the repeats is closely related to the frequency of recombination [63]; namely, the frequency of recombination mediated by short repeats tends to be lower than that mediated by long repeats. For example, in *Nymphaea colorata* Peter [64], *Scutellaria tsinyunensis* C. Y. Wu et S. Chow [65] and *Abelmoschus esculentus* (L.) Moench [66], the long repeats had high recombination frequencies, and the short repeats had lower recombination frequencies; all of the repeats identified in the *Prunus salicina* Lindl. [67] mitogenome were short repeats, and they all had low recombination frequency. In the *Q. acutissima* mitogenome, we found two pairs of long repeats that also had a high frequency of recombination, but some potential repeats involved in recombination have not been discovered.

During mitochondrial evolution, some chloroplast fragments migrate into the mitogenome, and the length and sequence similarity of the migrated fragments vary between species [12]. We found 23 fragments that were homologous to the chloroplast genome and mitogenome, with a total of 13 complete, 2 PCGs, 10 tRNA, and 1 rRNA genes. The transfer of tRNA genes from chloroplasts to mitochondria is common in angiosperms [68]. 

In plants, RNA editing is required for gene expression, and cytidine (C)-to-uridine (U) RNA editing is enriched in mitochondrial and chloroplast genomes [56]. Studying RNA editing sites helps to understand the expression of mitochondrial and chloroplast genes in plants. Previous research had uncovered approximately 491 RNA editing sites within 34 genes in *Oryza sativa* L. [48], 486 within 31 genes in *Phaseolus vulgaris* L. [14], and 421 within 26 genes in *Acer truncatum* Bunge [56]. In this study, 466 RNA editing sites were identified in 32 PCGs based on online site prediction, all of which exhibited C-U RNA editing. The number of RNA editing sites varied greatly among genes, but the largest numbers of cytochrome c biogenesis and NADH dehydrogenase genes were similar to those of *A**. truncatum* and *P**. vulgaris*. The identification of RNA editing sites provides clues to predicting the gene function of new codons.

The covariance study is the arrangement of homologous genes or sequences, and the results showed that a large number of homologous covariance blocks were detected between *Q. acutissima* and *F**. sylvatica*, *J**. mandshurica* and *A**. thaliana*, among which the largest covariance block with *F**. sylvatica* was nearly 20 kb, while the larger covariance blocks with *J**. mandshurica* and *A**. thaliana* were less than 20 kb. The inconsistent order of the co-linear blocks’ arrangement suggests that *Q. acutissima* mitogenome has undergone extensive genomic rearrangements from them and is extremely unconserved in structure, which should be the main reason for the evolution and diversification of *Q. acutissima* mitogenome. The *Q. acutissima* chloroplast genome was shown to be more conservative with similar characteristics to other genus *Quercus* species, and analysis of the phylogenetic relationships was found *Q. acutissima* to be closely related to *Q. variabilis* [25]. In this study, we further analyzed the phylogenetic relationships of *Q. acutissima* based on mitochondrial genomic information and constructed a sequence phylogenetic tree using PCGs. The results indicated that the affinity between *Q. acutissima* and *F**. sylvatica* is closer.

## 5. Conclusions

Here, we assembled and annotated the *Q. acutissima* mitogenome and performed extensive analyses based on DNA and amino acid sequences of annotated genes. The total length of *Q. acutissima* mitogenome is 448,982 bp, with 45.72% GC content. The mitogenome main structure is branched, including linear molecule 1, circular molecule 2, and circular molecule 3, but whether these molecules coexist needs further study. We annotated 32 PCGs, including 24 unique mitochondrial core and 8 non-core, 16 tRNA, and 3 rRNA genes. Additionally, the codon usage, repeated sequences, genome recombination, chloroplast to mitochondrion DNA transformation, RNA editing, and synteny were also analyzed. Phylogenetic trees based on the mitogenomes of 29 species contributed to the scientific classification of *Q. acutissima*. This study provided information on the genetic characteristics, phylogenetic relationships, and evolution of *Q. acutissima* as well as serves as a basis for species identification and biological research in Fagaceae.

## Figures and Tables

**Figure 1 genes-13-01321-f001:**
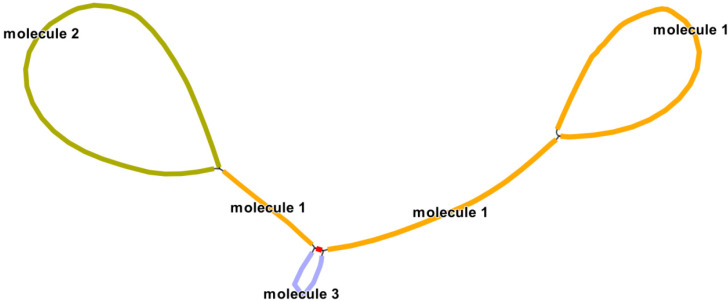
Branched conformation of *Q. acutissima* mitogenome.

**Figure 2 genes-13-01321-f002:**
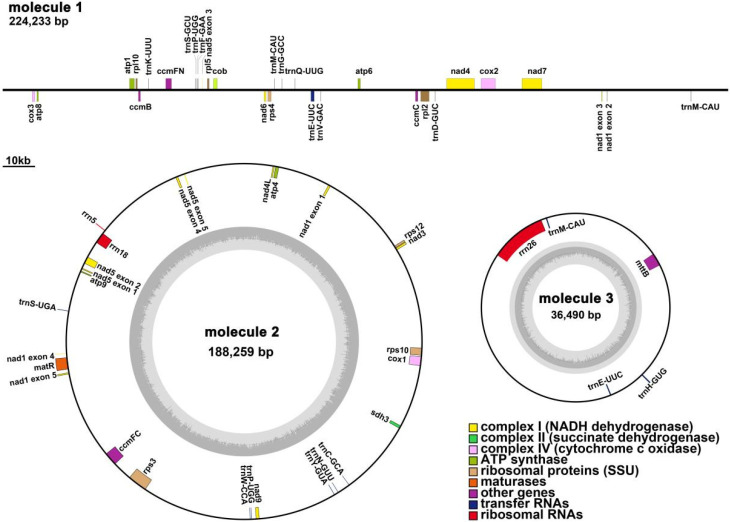
*Q. acutissima* mitogenome gene map. Genes shown on the outside and inside of the circle are transcribed clockwise and counterclockwise, respectively. The dark gray region in the inner circle depicts GC content.

**Figure 3 genes-13-01321-f003:**
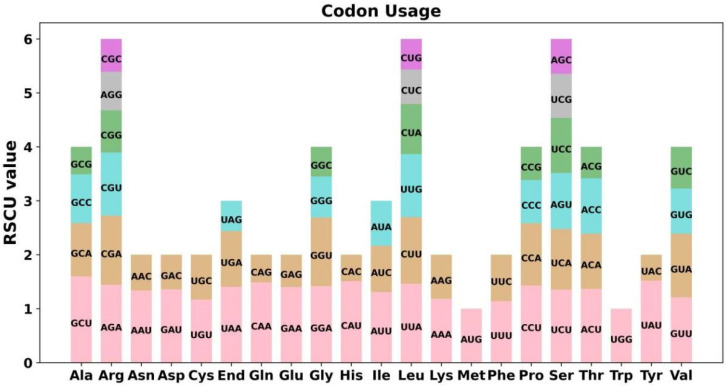
*Q. acutissima* mitogenome relative synonymous codon usage (RSCU). Codon families are shown on the *x*-axis. RSCU values are the number of times a particular codon is observed relative to the number of times that codon would be expected for a uniform synonymous codon usage.

**Figure 4 genes-13-01321-f004:**
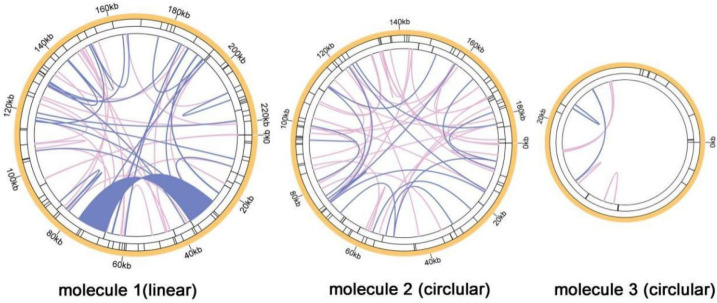
*Q. acutissima* mitogenome repeated sequence diagram. The colored line on the innermost circle connects two repeated sequences of interspersed repeats. The black line segments on the second circle and on the outermost circle denote tandem repeat and SSR, respectively.

**Figure 5 genes-13-01321-f005:**
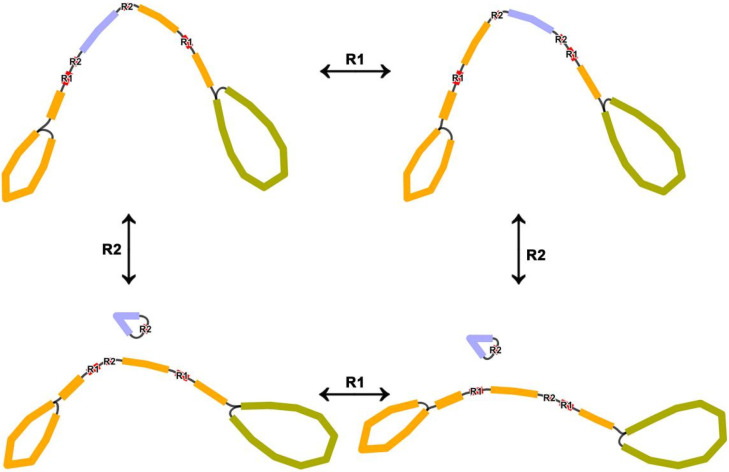
Two pairs of long repeated sequences are involved in mediating *Q.*
*acutissima* mitogenome reorganization. The orange node represents molecule 1, the yellow–green contig represents molecule 2, and the blue node represents molecule 3.

**Figure 6 genes-13-01321-f006:**
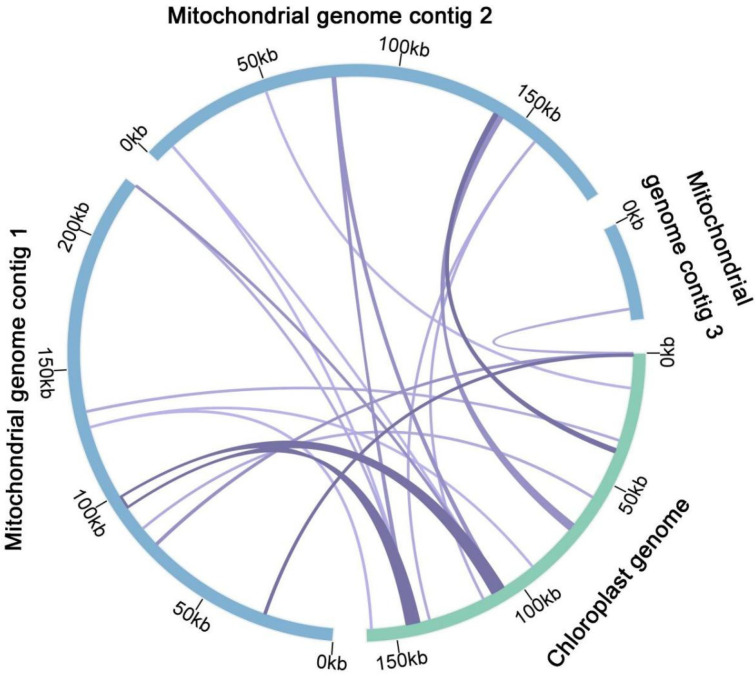
Schematic representation of gene transfers between chloroplast and mitogenomes in *Q. acutissima*. The blue and green arcs represent the mitogenome and chloroplast genome, respectively, and the purple lines between the arcs correspond to genomic fragments that are homologous.

**Figure 7 genes-13-01321-f007:**
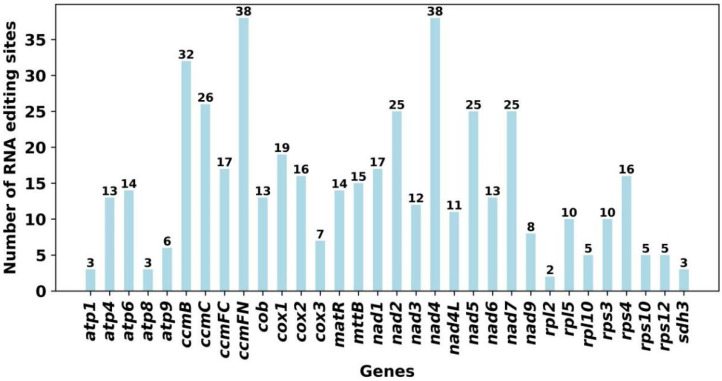
Number of RNA editing sites predicted by individual PCGs in mitochondria.

**Figure 8 genes-13-01321-f008:**
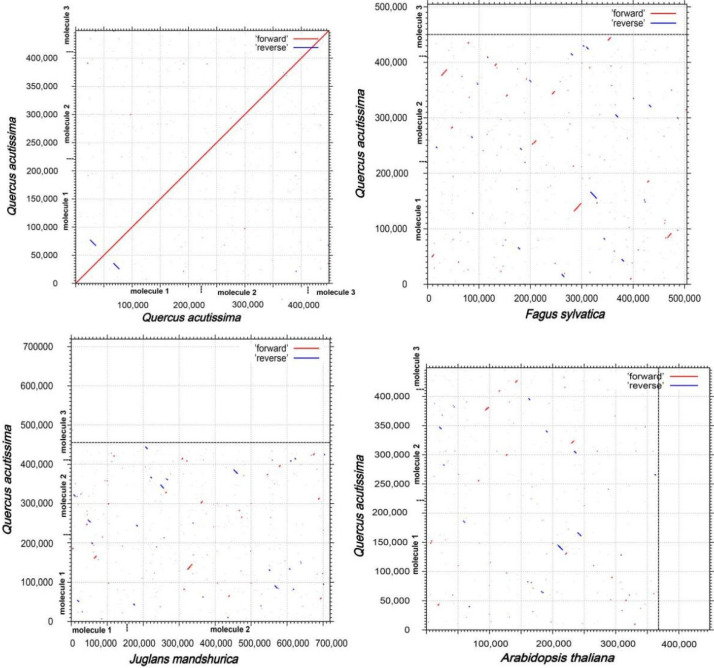
Dot-plots analysis. Self-dot plot of *Q. acutissima* is shown in the upper left corner, and the rest are dot plots of *Q. acutissima* with *F**. sylvatica*, *J**. mandshurica*, and *A**. thaliana*.

**Figure 9 genes-13-01321-f009:**
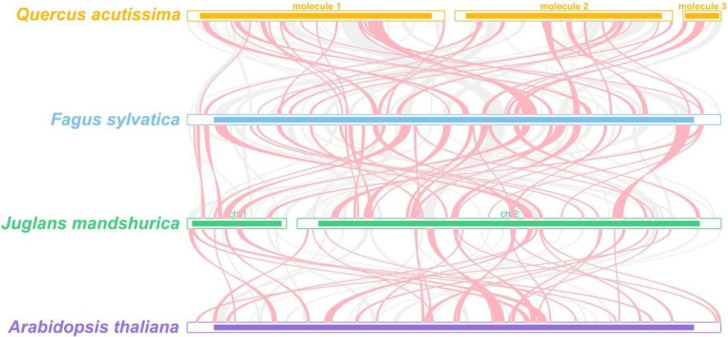
Mitogenome synteny. Bars indicated the mitogenomes, and the ribbons showed the homologous sequences between the adjacent species. The red areas indicate where the inversion occurred, the gray areas indicate regions of good homology. Common blocks less than 0.5 kb in length are not retained, and regions that fail to have a common block indicate that they are unique to the species. *A**. thaliana* was also compared.

**Figure 10 genes-13-01321-f010:**
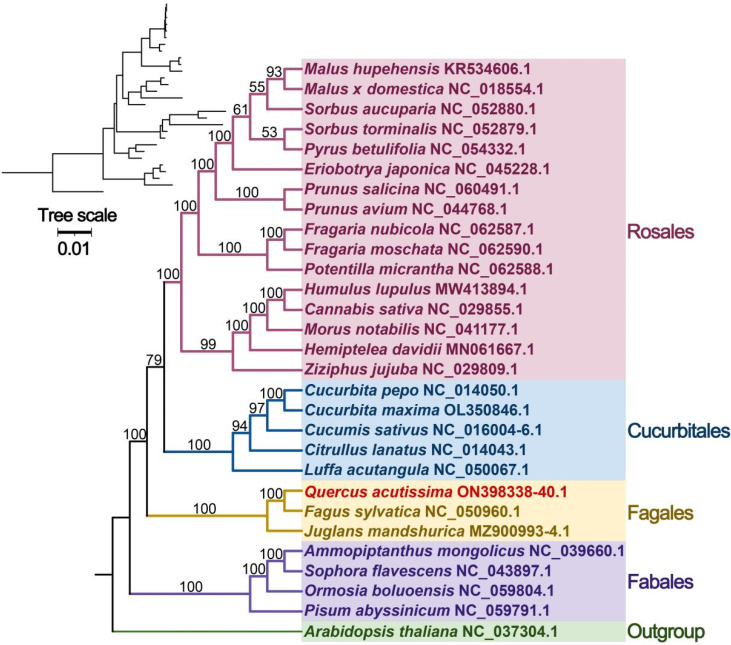
Phylogenetic tree of 29 angiosperms based on the sequences of 14 conserved mitochondrial PCGs. The *A**. thaliana* was chosen as the outgroup. The number at each node is the bootstrap probability.

**Table 1 genes-13-01321-t001:** Recombination frequency of the *Q**. acutissima* mitogenome related to two long repeats.

Repeat	Length (bp)	Location	Reads Support Major Conformation	Reads Support Alternative Conformation
R1	10,578	molecule 1: 35,975–25,398	19	8
		molecule 1: 66,867–77,444		
R2	1679	molecule 1: 68,252–66,574	224	182
		molecule 2: 34,939–36,490; 1–127		

**Table 2 genes-13-01321-t002:** Fragments transferred from chloroplasts to mitochondria in *Q. acutissima*.

	Alignment Length	Identity%	Mis-match	Gap Openings	CP Start	CP End	Mt Start	Mt End	MTPT Annotation
1	4760	99.979	1	0	106,031	110,790	98,643	93,884	Complete (*trn*V-GAC, *rrn*16S, *trn*I-GAU, *trn*A-UGC), Partial (*rrn*23S)
2	4760	99.979	1	0	140,802	145,561	93,884	98,643	Partial (*rrn*23S), Complete (*trn*A-UGC, *trn*I-GAU, *rrn*16S, *trn*V-GAC)
3	354	97.74	8	0	620	973	26,955	26,602	Partial (*psb*A)
4	354	97.74	8	0	620	973	75,887	76,240	Partial (*psb*A)
5	127	100	0	0	108,462	108,588	224,233	224,107	Partial (*trn*I-GAU)
6	127	100	0	0	143,004	143,130	224,107	224,233	Partial (*trn*I-GAU)
7	77	98.701	1	0	33,701	33,777	133,648	133,724	Complete (*trn*D-GUC)
8	75	94.667	4	0	57,523	57,597	83,943	84,017	Complete (*trn*M-CAU)
9	77	89.61	6	2	159,056	159,130	127,587	127,511	Complete (*trn*I-CAU)
10	77	89.61	6	2	92,462	92,536	127,511	127,587	Complete (*trn*I-CAU)
11	28	100	0	0	1015	1042	26,579	26,606	Partial (*psb*A)
12	28	100	0	0	1015	1042	76,263	76,236	Partial (*psb*A)
13	1149	99.739	3	0	37,300	38,448	140,389	139,241	Partial (*psb*D, *psb*C)
14	1017	77.778	150	46	71,581	72,585	141,644	142,596	Complete (*pet*L, *pet*G, *trn*W-CCA, *trn*P-UGG)
15	495	83.03	73	8	70,186	70,678	140,403	140,888	Partial (*psb*E)
16	889	73.903	177	42	106,819	107,682	76,204	75,346	Partial (*rrn*16S)
17	889	73.903	177	42	143,910	144,773	75,346	76,204	Partial (*rrn*16S)
18	83	98.795	1	0	136,687	136,769	157,954	157,872	Complete (*trn*N-GUU)
19	83	98.795	1	0	114,823	114,905	157,872	157,954	Complete (*trn*N-GUU)
20	55	96.364	2	0	143,130	143,184	8165	8111	Partial (*trn*I-GAU)
21	55	96.364	2	0	108,408	108,462	8111	8165	Partial (*trn*I-GAU)
22	38	100	0	0	13,356	13,393	49,527	49,490	Partial (*atp*A)
23	91	96.703	2	1	14	103	31,975	31,885	Complete (*trn*H-GUG)
Total	15,688								

## Data Availability

Publicly available datasets were analyzed in this study. These data can be found here: GenBank Accession Nos. MZ636519.

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
