# Peer review of "Complex Physical Structure of Complete Mitochondrial Genome of Quercus acutissima (Fagaceae): A Significant Energy Plant"

_genes, 2022, doi:10.3390/genes13081321_

Round 1
Reviewer 1 Report
The article “Complex physical structure of complete mitochondria genome of Quercus acutissima (Fagaceae): a significant energy plant” presents new and important data on the mitochondrial structure of Quercus acutissima and is therefore interesting to publish in the journal Genes. However, some changes in the article are necessary to improve its quality, especially in terms of structure and data presentation.
The Introduction develops the importance of studying this tree, briefly presenting the economic and environmental context for local communities. However, in the last paragraph of this topic it would be interesting to present the main objectives of this study in a summarized way. What are the main questions that are answered with this article? Attention, authors should not present any results in this topic.
All scientific names must be accompanied by the respective descriptor at least the first time they appear in the text. For example instead of “Quercus acutissima” it should appear “Quercus acutissima Carruth. “.
The materials and methods are well described and seem to me to be suitable for the type of study.
Line 128 – change the formatting of Q. acutissima to italics.
The results are well presented and do not raise major problems.
I believe that the title of Figure 2 “Quercus acutIssima mitochondrial genome” can be eliminated, as this information is already in the legend.
In the Discussion, the authors list and highlight the differences in the data obtained with other studies on species of the same family.
Line 298-299 – italicize “Acer truncatum”.
The Conclusion is succinct and adequate to the study, presenting the main results obtained in the analyzes carried out.
The bibliographic references are current and relevant to the subject of this study.
Author Response
Response to Reviewer 1 Comments
Point 1: The article “Complex physical structure of complete mitochondria genome of Quercus acutissima (Fagaceae): a significant energy plant” presents new and important data on the mitochondrial structure of Quercus acutissima and is therefore interesting to publish in the journal Genes. However, some changes in the article are necessary to improve its quality, especially in terms of structure and data presentation. The Introduction develops the importance of studying this tree, briefly presenting the economic and environmental context for local communities. However, in the last paragraph of this topic it would be interesting to present the main objectives of this study in a summarized way. What are the main questions that are answered with this article? Attention, authors should not present any results in this topic.
Response 1: According to the suggestion, in the last paragraph of this topic we have presented the main objectives of this study in a summarized way, while indicating the main questions answered in this paper. The revised text reads “Here, we assembled and annotated Q. acutissima mitogenome. This study amied to analyze relative synonymous codon usage (RSCU) and repeated sequences, detect genome recombination, assess gene transfer between chloroplast and mitochondrial genomes, and RNA editing sites, and explore synteny and phylogenetic relationships.”
Point 2: All scientific names must be accompanied by the respective descriptor at least the first time they appear in the text. For example instead of “Quercus acutissima” it should appear “Quercus acutissima Carruth. “. The materials and methods are well described and seem to me to be suitable for the type of study. Line 128 – change the formatting of Q. acutissima to italics. The results are well presented and do not raise major problems. I believe that the title of Figure 2 “Quercus acutIssima mitochondrial genome” can be eliminated, as this information is already in the legend.
Point 3: In the Discussion, the authors list and highlight the differences in the data obtained with other studies on species of the same family. Line 298-299 – italicize “Acer truncatum”. The Conclusion is succinct and adequate to the study, presenting the main results obtained in the analyzes carried out. The bibliographic references are current and relevant to the subject of this study.
Response 3: We have changed the formatting of Acer truncatum in line 298-299 to italics.

Reviewer 2 Report
The genus Quercus including over 500 species is widely distributed throughout the temperate regions of the Northern Hemisphere and, is a crop of major importance to forest industries such as timber, tanbark, or cork. These species play important roles in China’s forest ecosystem. The evolutionary histories of oaks are strongly influenced by palaeoenvironmental changes in different ways, such as historical introgression/hybridization triggered by range fluctuations, interspecific divergence along with the formation of geographical barriers, and lineage diversification caused by local adaptation to diverging climates.
Oak’s taxonomy, genetic structure, and breeding are complicated because of its wide variety of species, diverse forms, complex habitat conditions, and gene exchanges between species. The genus is characterized by high variability of morphological and ecological traits, the occurrence of mixed stands, the presence of large population sizes, and high levels of gene flow within the Quercus complex.
Quercus acutissima Carruthers, one of the members of section Cerris, is a dominant species in warm-temperate deciduous forests of East Asia. As an ecologically important and economically valuable tree species. Many studies have used nuclear simple sequence repeat (SSR) chloroplast DNA makers to study phylogeny and population variation. Recently published data on the chloroplast genome of Q. acutissima, but its mitogenome is still unexplored. Therefore, I believe that the study submitted for review, in terms of the selection of the research topic, is very important for the development of science.
This manuscript is in general well written, logically structured, well-illustrated, and easy to understand. It also addresses a subject that is of great interest in the scientific community. The title clearly describes the contents of the paper. The abstract is well written. It encapsulates the entire study. The introduction is well written as it gives a good background of the research in question. Also, the aim of the study is evident in the beginning and concluding parts. I believe that the Materials and Methods section is well structured and scientifically sound. The results are well presented, figures and tables are correct. Literature reviews in the discussion section of the manuscript are very professional.
I have no critical comments, in my opinion, the work may be published in its current version.
Minor:
Title: ‘mitochondria genome’ or ‘mitochondrial genome’?
Line 128: The Latin name Q. acutissima should be italicized
Author Response
Response to Reviewer 2 Comments
Point 1: The genus Quercus including over 500 species is widely distributed throughout the temperate regions of the Northern Hemisphere and, is a crop of major importance to forest industries such as timber, tanbark, or cork. These species play important roles in China’s forest ecosystem. The evolutionary histories of oaks are strongly influenced by palaeoenvironmental changes in different ways, such as historical introgression/hybridization triggered by range fluctuations, interspecific divergence along with the formation of geographical barriers, and lineage diversification caused by local adaptation to diverging climates.
Oak’s taxonomy, genetic structure, and breeding are complicated because of its wide variety of species, diverse forms, complex habitat conditions, and gene exchanges between species. The genus is characterized by high variability of morphological and ecological traits, the occurrence of mixed stands, the presence of large population sizes, and high levels of gene flow within the Quercus complex.
Quercus acutissima Carruthers, one of the members of section Cerris, is a dominant species in warm-temperate deciduous forests of East Asia. As an ecologically important and economically valuable tree species. Many studies have used nuclear simple sequence repeat (SSR) chloroplast DNA makers to study phylogeny and population variation. Recently published data on the chloroplast genome of Q. acutissima, but its mitogenome is still unexplored. Therefore, I believe that the study submitted for review, in terms of the selection of the research topic, is very important for the development of science.
This manuscript is in general well written, logically structured, well-illustrated, and easy to understand. It also addresses a subject that is of great interest in the scientific community. The title clearly describes the contents of the paper. The abstract is well written. It encapsulates the entire study. The introduction is well written as it gives a good background of the research in question. Also, the aim of the study is evident in the beginning and concluding parts. I believe that the Materials and Methods section is well structured and scientifically sound. The results are well presented, figures and tables are correct. Literature reviews in the discussion section of the manuscript are very professional.
I have no critical comments, in my opinion, the work may be published in its current version.
Minor:
Title: ‘mitochondria genome’ or ‘mitochondrial genome’?
Line 128: The Latin name Q. acutissima should be italicized
Response 1: We have changed ‘mitochondria genome’ of title to ‘mitochondrial genome’. And we have changed the formatting of Q. acutissima in line 128 to italics.

Round 2
Reviewer 1 Report
The authors read and accepted all the suggestions made, which resulted in a significant improvement in the quality of the article. Although the authors' responses were succinct, the text reflects all necessary improvements. Thus, as I have no further questions to ask, I consider that the article “Complex physical structure of complete mitochondrial genome of Quercus acutissima (Fagaceae): a significant energy plant” can continue with the process for publication in the journal Genes.